# Model-based Lifelong Reinforcement Learning with Bayesian Exploration

**Haotian Fu, Shangqun Yu, Michael Littman, George Konidaris**
Department of Computer Science, Brown University
{hfu7,syu68,mlittman,gdk}@cs.brown.edu

## Abstract

We propose a model-based lifelong reinforcement-learning approach that estimates a hierarchical Bayesian posterior distilling the common structure shared across different tasks. The learned posterior combined with a sample-based Bayesian exploration procedure increases the sample efficiency of learning across a family of related tasks. We first derive an analysis of the relationship between the sample complexity and the initialization quality of the posterior in the finite MDP setting. We next scale the approach to continuous-state domains by introducing a Variational Bayesian Lifelong Reinforcement Learning algorithm that can be combined with recent model-based deep RL methods, and that exhibits backward transfer. Experimental results on several challenging domains show that our algorithms achieve both better forward and backward transfer performance than state-of-the-art lifelong RL methods.[1]

## 1 Introduction

Reinforcement learning (RL) [42; 26] has been successfully applied to solve challenging individual tasks such as learning robotic control [11] and playing Go [38]. However, the typical RL setting assumes that the agent solves exactly one task, which it has the opportunity to interact with repeatedly. In many real-world settings, an agent instead experiences a collection of distinct tasks that arrive sequentially throughout its operational lifetime; learning each new task from scratch is inefficient, but treating them all as a single task will fail. Therefore, recent research has focused on algorithms that enable agents to learn across multiple, sequentially posed tasks, leveraging knowledge from previous tasks to accelerate the learning of new tasks. This problem setting is known as *lifelong reinforcement learning* [7; 50; 25]. The key questions in lifelong RL research are: How can an algorithm exploit knowledge gained from past tasks to quickly adapt to new tasks (forward transfer), and how can data from new tasks help the agent perform better on previously learned tasks (backward transfer)?

We propose to address these problems by extracting the common structure existing in previously encountered tasks so that the agent can quickly learn the dynamics specific to the new tasks. We consider lifelong RL problems that can be modeled as hidden-parameter MDPs or *HiP-MDPs* [10; 27], where variations among the true task dynamics can be described by a set of hidden parameters. Our algorithm goes further than previous work in both lifelong learning and HiP-MDPs by **1)** Separately modeling epistemic and aleatory uncertainty over different levels of abstraction across the collection of tasks: the uncertainty captured by a world-model distribution describing the probability distribution over tasks, and the uncertainty captured by a task-specific model of the (stochastic) dynamics within a single task. To enable more accurate sequential knowledge transfer, we separate the learning process for these two quantities and maintain a hierarchical Bayesian posterior that approximates them. **2)** Performing Bayesian exploration enabled by the hierarchical posterior: The method lets the agent act optimistically according to models sampled from the posterior, and thus increases sample efficiency.

---

[1]Code repository available at `https://github.com/Minusadd/VBLRL`.

36th Conference on Neural Information Processing Systems (NeurIPS 2022).

Specifically, we propose a model-based lifelong RL approach with Bayesian exploration that estimates a Bayesian world-model posterior that distills the common structure of previous tasks, and then uses this between-task posterior as a within-task prior to learn a task-specific model in each subsequent task. The learned hierarchical posterior model combined with sample-based Bayesian exploration procedures can increase the sample efficiency of learning. We first derive an explicit performance bound that shows that the task-specific model requires fewer samples to become accurate as the world-model posterior approaches the true underlying world-model distribution for the discrete case. We further develop Variational Bayesian exploration for Lifelong RL (VBLRL), a more scalable version that uses variational inference to approximate the distribution and leverages Bayesian Neural Networks (BNNs) [17; 3] to build the hierarchical Bayesian posterior. VBLRL provides a novel way to separately estimate different kinds of uncertainties in the HiP-MDP setting. Based on the same framework, we also propose a backward transfer version of VBLRL that is able to provide improvements for previously encountered tasks. Our experimental results on a set of challenging domains show that our algorithms achieve better both forward and backward transfer performance than state-of-the-art lifelong RL algorithms when given only limited interactions with each task.

## 2 Background

RL is the problem of maximizing the long-term expected reward of an agent interacting with an environment [42]. We usually model the environment as a Markov Decision Process or *MDP* [36], described by a five tuple: $\langle S, A, R, T, \gamma \rangle$, where $S$ is a finite set of states; $A$ is a finite set of actions; $R : S \times A \mapsto [0, 1]$ is a reward function, with a lower and upper bound of 0 and 1; $T : S \times A \mapsto \Pr(S)$ is a transition function, with $T(s'|s, a)$ denoting the probability of arriving in state $s' \in S$ after executing action $a \in A$ in state $s$; and $\gamma \in [0, 1)$ is a discount factor, expressing the agent's preference for delayed over immediate rewards.

An MDP is a suitable model for the task facing a single agent. In the lifelong RL setting, the agent instead faces a series of tasks $m_1, ..., m_n$, each of which can be modeled as an MDP: $\langle S^{(i)}, A^{(i)}, R^{(i)}, T^{(i)}, \gamma^{(i)} \rangle$. For lifelong RL problems, the performance of a specific algorithm is usually evaluated based on both forward transfer and backward transfer results [30]:

- *Forward transfer*: the influence that learning task $t$ has on the performance in future task $k \succ t$.

- *Backward transfer*: the influence that learning task $t$ has on the performance in earlier tasks $k \prec t$.

A key question in the lifelong setting is how the series of task MDPs are related; we model the collection of tasks as a HiP-MDP, where a family of tasks is generated by varying a latent task parameter $\omega$ drawn for each task according to the world-model distribution $P_\Omega$. Each setting of $\omega$ specifies a unique MDP, but the agent neither observes $\omega$ nor has access to the function that generates the task family. The dynamics $T(s'|s, a; \omega_i)$ and reward function $R(r|s, a; \omega_i)$ for task $i$ then depend on $\omega_i \in \Omega$, which is fixed for the duration of the task. The tasks are i.i.d. sampled from a fixed distribution and arrive one at a time.

## 3 Related work

The first category of lifelong RL algorithms learns a single model that encourages transfer across tasks by modifying objective functions. EWC [28] imposes a quadratic penalty that pulls each weight back towards its old values by an amount proportional to its importance for performance on previously-learned tasks to avoid forgetting. There are several extensions of this work based on the core idea of modifying the form of the penalty [29; 53; 33]. Another category of lifelong RL methods uses multiple models with shared parameters and task-specific parameters to avoid or alleviate the catastrophic problem [5; 24; 31]. The drawback of this method is that it is hard to incorporate the knowledge learned from previous tasks during initial training on a new task [31]. Nagabandi et al. [32] introduce a model-based continual learning framework based on MAML, but they focus on discovering when new tasks were encountered without access to task indicators.

Published HiP-MDP methods use Gaussian Processes [10] or Bayesian neural networks [27] to find a single model that works for all tasks, which may trigger catastrophic forgetting [5; 24]. Meta-RL [47; 12] and multi-task RL [35; 43] settings also attempt to accelerate learning by transferring knowledge from different tasks. Some work employs the MAML framework with Bayesian methods

to learn a stochastic distribution over initial parameters [51; 16; 13]. Other work uses the collected trajectories to infer the hidden parameter, which is taken as an additional input when computing the policy [37; 56; 14]. Our method, however, focuses on problems where the tasks arrive sequentially instead of having a large number of tasks available at the beginning of training. This sequential setting makes it hard to accurately infer the hidden parameters, but opens the door for algorithms that support backward transfer.

Some prior work uses Bayesian methods in RL to quantify uncertainty over initial MDP models [15; 1; 18]. Several algorithms start from the idea of sampling from a posterior over MDPs for Bayesian RL, maintaining Bayesian posteriors and sampling one complete MDP [41; 49] or multiple MDPs [2]. Instead of focusing on single-task RL, our algorithm aims to find a posterior over the common structure among multiple tasks. Wilson et al. [49] uses a hierarchical Bayesian infinite mixture model to learn a strong prior that allows the agent to rapidly infer the characteristics of a new environment based on previous tasks. However, it only infers the category label of a new MDP and only works in discrete settings.

# 4   Model-based Lifelong Reinforcement Learning

Our approach is built upon two main intuitions: First, transferring the transition model instead of policy/value function leads to more efficient usage of the data when "finetuning" on a new task. As we show empirically in Section 5.1, although some model-free lifelong RL algorithms perform better than the proposed model-based method in single-task cases, in lifelong RL setting of the same task type the model-based method is still able to achieve comparable/better performance with only half the amount of data. Secondly, with a model that is able to capture different levels of the uncertainty within HiP-MDPs, an agent can employ sample-based Bayesian exploration to further improve sample-efficiency.

The model underlying our approach is a hierarchical Bayesian posterior over task MDPs controlled by the hidden parameter $\omega$. Intuitively, we maintain probability distributions that separately capture two categories of uncertainty within lifelong learning tasks: The *world-model posterior* $P(\omega)$ captures the epistemic uncertainty of the world-model distribution over all future and past tasks $m_1, \cdots, m_n$ controlled by the hidden parameter $\omega_1, \cdots, \omega_n \sim P_\Omega$. As the learner is exposed to more and more tasks, this posterior should converge to the world-model distribution $P_\Omega$. The *task-specific posterior* $P(\omega_i)$ captures the epistemic uncertainty of the current task $m_i$ (Throughout the paper we will often write $i$ for simplicity.). As the learner is exposed to more and more transitions within the task, this posterior should approach the true distribution corresponding to $\omega_i$, i.e. peaking at the true $\omega_i$ for this specific task $i$, leaving only the aleatoric uncertainty of transitions within the task, which is independent of other tasks. Each time the agent encounters a new task, we initialize the task-specific model using the world-model posterior and further train it with data collected only from the new task. One of our key insights here is that the sample complexity of learning a new task will decrease as the initial prior of task-specific model approaches the true underlying distribution of the transition function. Thus, the agent can learn new tasks faster by exploiting knowledge common to previous tasks, thereby exhibiting positive forward transfer.

Specifically, we model the task-specific posterior via the transition dynamics using $p(s^{t+1}, r^t | s^t, a^t; \omega_i)$. The task-specific posterior, given a new state–action pair from task $i$, can be rewritten via Bayes' rule:

$$P(\omega_i | D_i^t, a^t, s^{t+1}, r^t) = \frac{P(\omega_i | D_i^t) P(s^{t+1}, r^t | D_i^t, a^t; \omega_i)}{P(s^{t+1}, r^t | D_i^t, a^t)}, \tag{1}$$

where $D_i^t = \{s^1, a^1, \cdots, s^t\}$ is the agent's history of task $i$ until time step $t$. The world-model posterior, given the new data from task $i$, can be rewritten as:

$$P(\omega | D_{1:i}) = \frac{P(\omega | D_{1:i-1}) P(D_i | D_{1:i-1}; \omega)}{P(D_i | D_{1:i-1})}, \tag{2}$$

where $D_{1:i}$ denotes the agent's history with all the experienced tasks $1 \sim i$ until current task $i$. In particular, each time when the agent faces a new task $i$ and has not started updating its task specific posterior yet (that is, $D_i^t = \varnothing$), we first use the world-model posterior to initialize the task-specific prior: $P(\omega_i | D_i^t) = P(\omega | D_{1:i})$. The world-model distribution aims to approximate the underlying

$P_\Omega$. The task-specific distribution aims to approximate the distribution that peaks at the true $\omega_i$ for this specific task $i$.

In Section 4.1, we derive a sample complexity bound in the finite MDP case which explicitly show how the distance between the distribution of a task's true transition model and our task-specific model's prior initialized by the parameters of the world-model posterior will affect the learning efficiency of a new task. Then, in Section 4.2 and 4.3 we extend our high-level idea to a scalable version that can be combined with recent model-based RL approaches and exhibits positive forward & backward transfer.

## 4.1 Sample Complexity Analysis

In this subsection, we consider the finite MDP setting and use a standard Bayesian exploration algorithm BOSS [2] as a single-task baseline. Note that BOSS creates optimism in the face of uncertainty as the agent can choose actions based on the highest performing transition of the $K$ models sampled, which drives exploration. We included explanations of the finite MDP version of our algorithm BLRL (Bayesian Lifelong RL) based on BOSS in appendix C and simple experiments on Gridworlds in appendix J.

BLRL uses the world-model posterior $P(\omega|D_{1:i-1})$ learned from previous $i-1$ tasks to initialize (copy distribution) the task-specific prior of $P(\omega_i)$ of new task $i$, aiming to decrease the number of samples needed to learn an accurate task-specific posterior. Our analysis focuses on how the properties of the Bayesian prior affect the sample complexity of learning each specific task.

Let $\pi(\omega_i)$ denote the prior distribution on the parameter space $\Gamma$. We consider a set of transition-probability densities $p(\cdot|\omega_i) = p(s^{t+1}, r^t|s^t, a^t, \omega_i)$ indexed by $\omega_i$, and the true underlying density $q$. We also denote the model family $\{p(\cdot|\omega_i) : \omega_i \in \Gamma\}$ by the same symbol $\Gamma$.

**Lemma 4.1.** *The task-specific posterior in Equation 1 can be regarded as the probability density $g(\omega_i)$ with respect to $\pi$ that attains the infimum of:*

$$R_n(g) = \mathbb{E}_\pi g(\omega_i) \sum_{t=1}^{T} \ln \frac{q(s^{t+1}, r^t|D_i^t, a^t)}{p(s^{t+1}, r^t|D_i^t, a^t; \omega_i)} + D_{KL}(gd\pi||d\pi). \qquad (3)$$

$\inf R_n(g)$ controls the complexity of the density estimation process for $g(\omega_i)$. Intuitively, Lemma 4.1 converts the Bayesian posterior into an information theoretical minimizer that allows us to further investigate the relationship between the properties of the Bayesian prior and the risk/complexity of attaining the posterior.

**Proposition 4.2.** *Define the **prior-mass radius** of the transition-probability densities as:*

$$d_\pi = \inf\{d : d \geq -\ln \pi(\{p \in \Gamma : D_{KL}(q||p) \leq d\})\}. \qquad (4)$$

*Intuitively, this quantity measures the distance between the Bayesian prior of the task-specific model and the true underlying task-specific distribution. Then, we adopt the same settings in Zhang [55]: $\forall \rho \in (0,1)$ and $\eta \geq 1$, let*

$$\varepsilon_n = (1 + \frac{1}{n})\eta d_\pi + (\eta - \rho)\varepsilon_{upper,n}((\eta-1)/(\eta-\rho)), \qquad (5)$$

*where $\varepsilon_{upper,n}$ is the **critical upper-bracketing radius** [55]. The decay rate of $\varepsilon_{upper,n}$ controls the consistency of the Bayesian posterior distribution [2]. Let $\rho = \frac{1}{2}$, we have for all states and actions, $h \geq 0$ and $\delta \in (0,1)$, with probability at least $1 - \delta$,*

$$\pi_n\Big(\Big\{p \in \Gamma : ||p - q||_1^2/2 \geq \frac{2\varepsilon_n + (4\eta - 2)h}{\delta/4}\Big\}\Big|X\Big) \leq \frac{1}{1 + e^{nh}}, \qquad (6)$$

*where $\pi_n(\omega_i)$ denotes the posterior distribution over $\omega$ after collecting $n$ samples, $X$ denotes the $n$ collected samples. It functions the same as $D_i^t$ in Equation 1.*

Similar to BOSS, for a new MDP $m^* \sim M$ with hidden parameters $\omega_{m^*}$, we can define the Bayesian concentration sample complexity for the task-specific posterior: $f(s, a, \epsilon_0, \delta_0, \rho_0)$, as the minimum number $u$ such that, if $u$ IID transitions from $(s, a)$ are observed, then, with probability at least $1 - \delta_0$,

$$Pr_{m \sim posterior}(||T(\cdot|s, a, \omega_m) - T(\cdot|s, a, \omega_{m^*})||_1 < \epsilon_0) \geq 1 - \rho_0. \qquad (7)$$

Intuitively, the inequality means that for a model with the hidden parameter sampled from the learned posterior distribution, the probability that that it is within $\epsilon_0$ far away from the true model is larger than $1 - \rho_0$.

**Lemma 4.3.** *Assume the posterior of each task is consistent (that is, $\varepsilon_{upper,n} = o(1)$) and set $\eta = 2$, then the Bayesian concentration sample complexity for the task-specific posterior $f(s, a, \epsilon, \delta, \rho) = O\left(\frac{d_\pi + \ln \frac{1}{\rho}}{\epsilon^2 \delta - d_\pi}\right)$.*

*Proof (sketch).* This bound can be derived by directly combining Lemma 4.2 and Equation 7. □

The above lemma suggests an upper bound of the Bayesian concentration sample complexity using the prior-mass radius. We can further combine this result with PAC-MDP theory [39] and derive the sample complexity of the algorithm for each new task.

**Proposition 4.4.** *For each new task, set the sample size $K = \Theta(\frac{S^2 A}{\delta} \ln \frac{SA}{\delta})$ and the parameters $\epsilon_0 = \epsilon(1 - \gamma)^2, \delta_0 = \frac{\delta}{SA}, \rho_0 = \frac{\delta}{S^2 A^2 K}$, then, with probability at least $1 - 4\delta$, $V^t(s^t) \geq V^*(s^t) - 4\epsilon_0$ in all but $\tilde{O}(\frac{S^2 A^2 d_\pi}{\delta \epsilon^3 (1 - \gamma)^6})$ steps, where $\tilde{O}(\cdot)$ suppresses logarithmic dependence.*

*Proof (sketch).* The central part of the proof is Proposition 4.2 (detailed proof in appendix), and the re- maining parts are exactly the same as those for BOSS. In general, the proof is based on the PAC-MDP theorem [40] combined with the new bound for the Bayesian concentration sample complexity we derived in Lemma 2. For each new task, the main difference between BLRL and BOSS is that we use the world-model posterior to initialize the task-specific posterior, which results in a new sample complexity bound with $d_\pi$. □

The result formalizes how the sample complexity of the lifelong RL algorithm will change with respect to the initialization quality of the posterior: if we put a larger prior mass at a density close to the true $q$ such that $d_\pi$ is small, the sample efficiency of the algorithm will increase accordingly. In other words, the sample complexity of our algorithm drops proportionally to $d_\pi$, which is the distance between the Bayesian prior of the task-specific model initialized by the parameters of the world-model posterior and the true underlying task-specific distribution. We provide a illustrative example in the appendix O.

## 4.2 Variational Bayesian Lifelong RL

The intuition from the last section is that, if we initialize the task-specific distribution with a prior that is close to the true distribution, sample complexity will decrease accordingly. To scale our approach, we must find a efficient way to explicitly approximate these distributions. We propose a practical approximate algorithm, VBLRL, that uses neural networks and variational inference [21].

We choose Bayesian neural networks (BNN) to approximate the posterior. The intuition is that, in the context of stochastic outputs, BNNs naturally approximate the hierarchical Bayesian model since they also maintain a learnable distribution over their weights and biases [17; 23]. We use the uncertainty embedded in the weights and biases of networks to capture the epistemic uncertainty introduced by hidden parameters of different tasks, while we also set the outputs of the neural networks to be stochastic to capture the aleatory uncertainty within each specific task. In our case, the BNN weights and biases distribution $q(\omega; \phi)$ (a distribution over $\omega$ but parameterized by $\phi$) can be modeled as fully factorized Gaussian distributions [3]:

$$q(\omega; \phi) = \prod_{j=1}^{|\Omega|} \mathcal{N}(\omega_j | \mu_j, \sigma_j^2), \tag{8}$$

where $\phi = \{\mu, \sigma\}$, and $\mu$ is the Gaussian's mean vector while $\sigma$ is the covariance matrix diagonal.

We maintain a world-model BNN across all the tasks and a task-specific BNN for each task. The input for all the BNNs is a state–action pair, and the output are the mean and variance of the prediction for reward and next state. Then, the posterior distribution over the model parameters can be computed leveraging variational lower bounds [22; 23]:

$$\phi_t = \arg\min_{\phi} \left[ D_{KL}[q(\omega; \phi) || p(\omega)] - \mathbb{E}_{\omega \sim q(\cdot; \phi)}[\log p(s^{t+1}, r^t | D^t, a^t; \omega)] \right], \tag{9}$$

where $p(\omega)$ represents the fixed prior distribution of $\omega$, also recall that $D^t = \{s^1, a^1, r^1, \cdots, s^t\}$. In VBLRL, $p(\omega)$ for the task-specific distribution is initialized by the world-model distribution, while $p(\omega)$ of the world-model distribution is simply set to a Gaussian. (That could be improved with more informed task family knowledge.) The second term on the right hand side can be approximated through $\frac{1}{N} \sum_{i=1}^{N} \log p(s^{t+1}, r^t | D^t, a^t; \omega_i)$ with $N$ samples from $\omega_i \sim q(\phi)$. This optimization can be performed in parallel for each $s$, keeping $\phi_{t-1}$ fixed. Details about our BNN structure can be found in appendix B.

Once we have the world-model as well as the task-specific posterior model, we can do posterior sampling to drive Bayesian exploration. Osband et al. [34] and Rakelly et al. [37] already show the benefit of posterior sampling for improving sample efficiency in single-task RL and meta-RL settings. Here, instead of sampling from the posterior of value functions or latent context representations, we directly sample from the posterior of the transition model and choose the optimal action based on the transition samples. Then we use the collected samples to update the posterior and do sampling again. The agent thus continually do Bayesian exploration and acts more and more optimally as the posterior distribution narrows.

We provide the framework of VBLRL in Figure 1. The outer loop (yellow rectangle) stands for the loop of training world-model BNN across tasks (noted at the bottom-right corner), the inner loop (cyan rectangle) stands for the training of the task-specific BNN within each specific task. We employ our posterior knowledge models in the context of a model-based RL method. When encountering a new task, VBLRL first uses the model parameters (that is, $\{\mu, \sigma\}$ of weights and biases of BNN) from the general knowledge model to initialize the task-specific posterior network (directly copy parameters). We sample transitions from the task-specific posterior and select actions based on the generated transitions. The detailed algorithm is summarized in Algorithm 1. Specifically, for planning, at each step we begin by creating $P$ particles from the current state $s^p_{\tau=t} = s_t \forall p$. Then, we sample $N$ candidate action sequences $a_{t:t+T}$

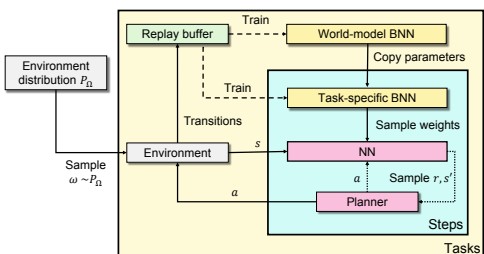

Figure 1: Variational Bayesian Lifelong RL (VBLRL). "NN" is the network using fixed parameters sampled from the weight distribution of the task-specific BNN. When the agent does CEM planning, we let it propagate the state particles using several NNs that use different network parameters, all sampled from the BNN to achieve randomness in transitions.

from a learnable distribution. These two steps are the same as PETS [8]. Then we propagate the state–action pairs using the learned task-specific model $p_{m_i}(\cdot|s, a)$ (BNN) and use the cross entropy method [4] to update the sampling distribution to make the sampled action sequences close to previous action sequences that achieved high reward. We further calculate the cumulative reward estimated (via the learned model) for previously sampled sequences and select the current action based on the mean of that distribution. By sampling from the task-specific posterior at each step, the agent explores in a temporally extended and diverse manner.

For each specific task, we use the task-specific BNN to plan during forward learning. They are updated with only the data from the current task thus avoid catastrophic forgetting in the forward transfer process. The world model is updated using the data from **all** the visited tasks. The intuition is to guide the two posteriors to separately learn two categories of uncertainty within lifelong learning tasks. The world-model posterior captures the epistemic uncertainty of the general knowledge distribution (shared across all tasks controlled by the hidden parameters) via the internal variance of world-model BNN. As the learner is exposed to more and more tasks, the posterior should converge to $P_\Omega$. The task-specific posterior captures the epistemic uncertainty of the current task $m_i$, which comes from the aleatory uncertainty of the world model when generating $\omega_i$ for a new task, via the internal variance of task-specific BNN. We provide additional illustration in Figure 9. In general, we used the world-model BNN to encode $P_\Omega$ over a family of tasks, and the task-specific BNN to encode $P(\omega_i)$, which should peak at the specific true $\omega_i$ sampled for the current task.

Note that in single-task cases, other CEM-based algorithms like PETS usually maintain a set of neural networks using the same training data, and sample action sequences from each of the neural nets to achieve randomness in transitions. However, in lifelong RL settings, it is unrealistic to maintain

($\geq 30$) models for each task encountered. Our usage of BNNs avoids such problems as we only have to train one neural network using the same data for each task, and we can sample an unlimited number of different action sequences to cover more possibilities as needed. In PETS, the epistemic uncertainty is estimated via the variance of the output mean of different neural networks, while, in VBLRL, it is estimated via the variance of the weights and biases distribution of the BNN.

### 4.3 Backward Transfer of Variational Bayesian Lifelong RL

In our lifelong RL setting, the agent interacts with each task for only a limited number of episodes and the task-specific model stops learning when the next task is initiated. As a result, there may exist portions of the transition dynamics in which model uncertainty remains high. However, as the world-model posterior continues to train on new tasks, it gathers more experience in the whole state space and can provide improved guesses concerning the "unknown" transition dynamics, even for previously encountered tasks.

Intuitively, the performance of an agent on one task has the potential to be further improved (positive backward transfer) if there exists a sufficiently large set of state–action transition pairs of which the task-specific model's predictions are not confident due to lack of data. In our algorithm, the aleatory uncertainty (irreducible chance in the outcome) is measured by the output variance of the prediction $\{\sigma_{r_\tau^p}, \sigma_{s_\tau^p}\}$, and the epistemic uncertainty (due to lack of experience) corresponds to the uncertainty of the output mean and variance (see Definition 1 below). Thus, a straightforward method to improve a previously learned task-specific model is to find the predictions it needs to make that have high epistemic uncertainty, and replace them with the predictions from the world-model posterior, which has lower epistemic uncertainty. If we only consider reward prediction, the quantity for measuring whether a task-specific/world model is sufficiently confident is as follows.

**Definition 4.5.** For a given state–action pair $(s, a)$, we define the confidence level $c$ of the predictions (reward) $r_\tau^p(s, a)$ from a task-specific/world model as:

$$c = -\frac{\sum_{p=1}^{P}(\mu_{r_\tau^p} - \overline{\mu}_{r_\tau^p})^2}{P - 1} - \alpha * \frac{\sum_{p=1}^{P}(\sigma_{r_\tau^p} - \overline{\sigma}_{r_\tau^p})^2}{P - 1}, \tag{10}$$

where $P$ is the number of particles and $\alpha$ is a hyperparameter controlling the scale of the second term. A similar definition applies to the task-specific/world model's next-state prediction. Intuitively, $c$ measures the uncertainty of the output mean and variance for each dynamic prediction. During CEM planning, we propagate each pair of $P$ state particles with different network parameters thanks to the usage of BNN, resulting in $P$ different $\{\mu_{r_\tau^p}, \sigma_{r_\tau^p}\}$. Thus we can further calculate the confidence level of current predictions based on these $P$ pairs of output mean and variance.

We show the detailed backward transfer algorithm in Algorithm 2. Compared with the forward training version, the agent will also calculate the confidence level of the task-specific and general-knowledge model for each particular transition and compare the value of them when predicting the state particles. If the task-specific model is not confident enough for this state–action pair (i.e. $c_{m_i} < c_{wm}$), we will use the world-model to do the predictions instead.

## 5 Experiments

### 5.1 OpenAI Gym MuJoCo Domains

We evaluated the performance of VBLRL on HiP-MDP versions of several continuous control tasks from the Mujoco physics simulator [45], *HalfCheetah-gravity, HalfCheetah-bodyparts, Hopper-gravity, Hopper-bodyparts, Walker-gravity, Walker-bodyparts*, all of which are lifelong-RL benchmarks used in prior work [31]. For each of six different domains, the task-specific hidden parameters correspond to different gravity values or different sizes and masses of the simulated body parts. Details can be found in the appendix. Compared with prior work, we substantially reduced the number of iterations that the agent can sample and train on: 100 iterations for each task and a horizon of 100 (Halfcheetah) or 400 (Hopper & Walker) for each iteration. We used such settings to increase the difficulty of lifelong learning and test the sample efficiency of lifelong RL algorithms, given that it is hard for a single-task training algorithm to obtain a good policy within such limited number of interactions with the environments. However, we show in appendix H and I that, as the agent is

exposed to more and more tasks, our lifelong learning agent is able to achieve similar performance to that of more fully trained single-task agents while requiring far fewer per-task samples.

| | VBLRL | T-HiP-MDP | LPG-FTW ($2\times$ samples) | EWC ($2\times$ samples) | Single-task MBRL |
|---|---|---|---|---|---|
| *CG*-Start | $\mathbf{160.68 \pm 48.80}$ | $126.95 \pm 31.41$ | $-81.59 \pm 9.18$ | $-3426.76 \pm 827.99$ | $-83.96 \pm 60.10$ |
| *CG*-Train | $\mathbf{226.72 \pm 26.53}$ | $170.20 \pm 39.92$ | $-29.49 \pm 11.03$ | $-3440.66 \pm 1007.50$ | $-40.47 \pm 10.68$ |
| *CG*-Back | $\mathbf{231.79 \pm 23.49}$ | $97.84 \pm 22.04$ | $-29.95 \pm 11.64$ | $-6672.33 \pm 3748.63$ | / |
| *CB*-Start | $\mathbf{110.74 \pm 41.96}$ | $78.95 \pm 18.43$ | $-263.94 \pm 40.80$ | $-5016.93 \pm 1708.10$ | $-101.02 \pm 39.11$ |
| *CB*-Train | $\mathbf{173.97 \pm 78.26}$ | $87.20 \pm 9.42$ | $-217.86 \pm 42.82$ | $-5454.52 \pm 2145.82$ | $-58.93 \pm 33.24$ |
| *CB*-Back | $\mathbf{181.60 \pm 67.50}$ | $116.03 \pm 17.35$ | $-116.41 \pm 65.64$ | $-13889.31 \pm 6851.05$ | / |
| *HG*-Start | $268.11 \pm 33.29$ | $230.21 \pm 30.75$ | $305.63 \pm 34.55$ | $\mathbf{306.79 \pm 29.14}$ | $18.62 \pm 1.51$ |
| *HG*-Train | $332.89 \pm 23.68$ | $285.87 \pm 41.48$ | $\mathbf{352.10 \pm 25.25}$ | $345.47 \pm 40.26$ | $20.46 \pm 1.77$ |
| *HG*-Back | $\mathbf{360.94 \pm 7.71}$ | $312.11 \pm 55.45$ | $347.07 \pm 44.09$ | $301.06 \pm 114.11$ | / |
| *HB*-Start | $193.27 \pm 8.03$ | $181.94 \pm 12.97$ | $\mathbf{256.13 \pm 69.68}$ | $133.95 \pm 27.61$ | $19.12 \pm 1.75$ |
| *HB*-Train | $\mathbf{296.34 \pm 11.15}$ | $227.50 \pm 41.78$ | $285.62 \pm 78.18$ | $139.01 \pm 46.24$ | $21.25 \pm 2.67$ |
| *HB*-Back | $\mathbf{316.74 \pm 20.72}$ | $213.93 \pm 75.95$ | $281.99 \pm 74.96$ | $-384.27 \pm 199.27$ | / |
| *WG*-Start | $\mathbf{248.97 \pm 19.95}$ | $217.94 \pm 39.99$ | $140.75 \pm 67.64$ | $163.61 \pm 86.55$ | $4.32 \pm 0.23$ |
| *WG*-Train | $\mathbf{333.65 \pm 26.73}$ | $268.97 \pm 36.26$ | $166.00 \pm 45.04$ | $181.76 \pm 97.42$ | $210.41 \pm 39.34$ |
| *WG*-Back | $\mathbf{364.18 \pm 49.60}$ | $173.33 \pm 58.43$ | $168.39 \pm 94.19$ | $216.09 \pm 62.67$ | / |
| *WB*-Start | $\mathbf{222.88 \pm 34.49}$ | $220.07 \pm 20.31$ | $164.79 \pm 10.31$ | $164.00 \pm 34.31$ | $4.64 \pm 0.23$ |
| *WB*-Train | $\mathbf{311.41 \pm 16.30}$ | $266.97 \pm 26.72$ | $165.75 \pm 2.39$ | $206.97 \pm 41.40$ | $196.42 \pm 22.04$ |
| *WB*-Back | $\mathbf{317.82 \pm 13.90}$ | $229.10 \pm 16.60$ | $195.01 \pm 49.25$ | $152.94 \pm 70.59$ | / |

Table 1: Results on OpenAI Gym Mujoco domains. *CG* denotes **Cheetah-Gravity**, *CB* denotes **Cheetah-Bodyparts**, *HG* denotes **Hopper-Gravity**, *HB* denotes **Hopper-Bodyparts**, *WG* denotes **Walker-Gravity**, *WB* denotes **Walker-Bodyparts**.

We compared VBLRL against: 1. state-of-the-art lifelong RL method LPG-FTW [31], 2. EWC [28], which is a single-model lifelong RL algorithm that achieves comparable performance with LPG-FTW as shown in the latter paper, 3. single-task Model-Based RL (using the same BNN structure and planning procedures), which stands for training the agent from scratch for each new task and do not use the world model to initialize the task-specific model, 4. T-HiP-MDP [27], which is a model-based lifelong RL baseline. For a fair comparison, we further replaced the DDQN algorithm [46] used in T-HiP-MDP with CEM planning, and let the transition model also predict the reward for each state–action pair. This modified baseline is similar to the single-model version of VBLRL (i.e., only using the world-model posterior across all the tasks). Moreover, LPG-FTW and EWC are built upon a relatively simple policy gradient baseline, which does not perform as well as our model-based baseline in the single-task setting within the same amount of interactions. Thus, we let them collect **twice the amount of samples** each iteration compared to VBLRL. We show in Appendix H that, with such modifications, in single-task settings the baselines achieve better or at least similar performance compared to the single-task version of VBLRL.

The results are shown in Table 1 and Figure 7 in Appendix I. For all six domains, we report the average performance of all the tasks at the beginning of training (**Start**) and after all training for each new task (**Train**), as well as the average performance for all previous tasks after training for a given number of tasks, which is our backward transfer test (**Back**). As shown in the results, our VBLRL shows better performance on all three test stages of the HalfCheetah domain and Walker domain, as well as better backward transfer performance on Hopper-gravity and Hopper-bodyparts than the other three algorithms. Comparing Figure 6 and Figure 7, we find that, in the Hopper domains, even though the single-task baseline used by LPG-FTW and EWC performed much better than VBLRL after we let them collect $\mathbf{2}\times$ samples each iteration, VBLRL still achieves comparable performance with LPG-FTW and EWC in lifelong RL experiments, suggesting that VBLRL is capable of making better use of the data in the lifelong learning setting. In the walker domains where the single-task baselines achieved similar performance, VBLRL shows significantly better performance than LPG-FTW/EWC in lifelong RL experiments. These comparisons also suggest that VBLRL's lifelong learning performance may be further improved when combined with better model-based deep RL algorithms, like Dreamer [20]. EWC fails in some of the tasks as it is hard to directly learn a single shared policy that achieves good performance when the tasks are highly diverse. T-HiP-MDP shows good results on some of the tasks because it is more sample-efficient in learning a shared model across all the tasks and more easily captures the world-model uncertainty. However, it cannot achieve as good performance as VBLRL as it is hard to model the task-specific uncertainty using only one model across all tasks, which also leads to negative backward transfer performance on tasks like Cheetah-Gravity. Comparing VBLRL's performance on the **Train** stage and **Back** stage, we also find that it shows positive backward transfer results on most tasks, without showing patterns of catastrophic forgetting. Overall, VBLRL's world-model posterior contributes to better forward transfer performance (**Start**), the learning of task-specific posterior contributes to

better forward transfer training for each new task (**Train**), and the combination of these two posteriors guides the agent to achieve better backward transfer performance (**Back**). Comparing our method to LPG-FTW and EWC highlights the advantage of model-based techniques for sample efficiency in lifelong RL setting, while comparing to T-HiP-MDP and single-task MBRL demonstrates the importance of estimating both global and local uncertainty.

## 5.2 Meta-world Domains

Meta-World [52] contains a suite of challenging evaluation benchmarks on a robotic SAWYER arm. We evaluated the performance of VBLRL as well as the baselines on two Meta-World task sets: Reach and Reach-Wall following the settings given by [54]. We used the v1 version of Meta-World, with state observations and dense versions of the reward functions. The hidden parameters in these cases are the goals we want the robot to reach, which controls the reward functions and are not included in the agent's observation space. We measured performance in terms of average cumulative rewards across tasks. As shown in Figure 2, VBLRL achieves significantly higher

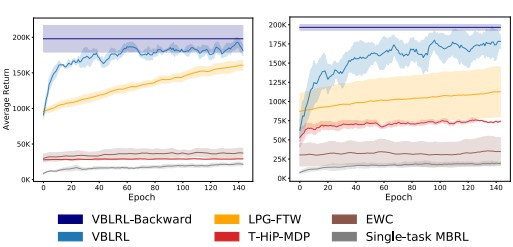

Figure 2: Average performance during training across all tasks. Left: Reach; Right: Reach-Wall.

performance than the other baselines. VBLRL-Backward denotes VBLRL's average backward transfer performance on all the tasks and is higher than the final performance of VBLRL's forward training in both tasks, suggesting that our approach can achieve positive backward transfer.

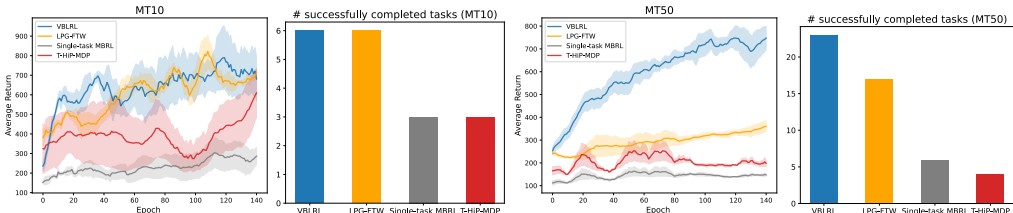

Figure 3: Performance comparison of VBLRL with the other baselines on MT10 and MT50. We also compare the **maximum** number of successfully completed tasks of different methods. The performance of VBLRL is similar to LPG-FTW on MT10 but significantly outperforms LPG-FTW on MT50 as the world-model posterior becomes more accurate when the number of tasks is larger.

We include additional experiments on MT10 and MT50 domains of MetaWorld (v2) in Figure 3. MT10 and MT50 consist of 10 and 50 different categories of manipulation tasks, including reach, push, drawer close, button press etc., and is a highly difficult setting for multi-task RL. We modified these two domains to be lifelong learning settings: instead of getting all 10/50 tasks at the same time at the beginning, we let them arrive sequentially. For instance, the first task is "reach" and we let the agent interact with it for 150 iterations, and then switch to "push" and the agent cannot collect data from "reach" anymore, after another 150 iterations the task changes to "pick-place", etc. We compare the results with LPG-FTW, T-HiP-MDP as well as Single-task MBRL (train the agent from scratch for each new task without using the world-model posterior to initialize the task specific model). With the limited number of interactions for each task, our proposed algorithm is still able to achieve reasonably good performance in such settings. In MT10, the total number of tasks (10) is relatively small so the world-model distribution may be far from accurate, but VBLRL still performs better than Single-task MBRL which does not use the world model to initialize the task-specific model. In MT50, the number of tasks is large enough (50) and we can see that VBLRL significantly outperforms all the other baselines. This is consistent with our intuition that sample efficiency improves in the task specific regime when the world model approaches the environment distribution. As the single-task model-based RL baseline we used here is relatively simple (CEM planning) to be comparable with the single-task baseline used by LPG-FTW, it is reasonable to expect the lifelong learning performance to be even better after combining our framework with model-based methods like Dreamer [20]; we leave such evaluations for future work.

### 5.3 Ablation Study

This section evaluates the essentiality of VBLRL's components. As shown in Figure 4 left, we tested different backward transfer strategy for VBLRL. Task-specific-Backward denotes the performance if directly using the learned task-specific model to do all the predictions without using world model. World-model-Backward denotes the performance if using world-model to do all the predictions without using each task-specific model. Compared to these baselines, VBLRL's backward transfer strategy achieves the best performance by combining these two kinds of models according to their confidence level. We also explored the in-

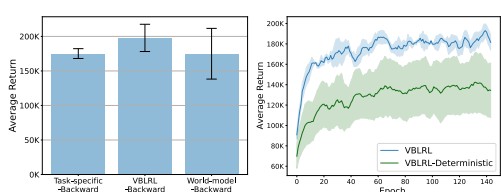

Figure 4: Ablation Study on Reach domain. Left: Backward transfer strategy; Right: Using Bayesian neural networks to model the task-specific posterior.

fluence of using a Bayesian neural network to model the task-specific model. As shown in the right side of the figure, using BNN enables faster adaptation to new tasks compared with using regular neural networks (VBLRL-Deterministic). We also include ablation studies on the number of particles in CEM planning in Appendix N.

## 6 Conclusion and Discussion

To improve sample efficiency in lifelong RL, our work proposed a model-based lifelong RL approach that distills shared knowledge from similar MDPs and maintains a Bayesian posterior to approximate the distribution derived from that knowledge. We gave a sample-complexity analysis of the algorithm in the finite MDP setting. Then, we extended our method to use variational inference, which scales better and supports both backward and forward transfer. Our experimental results show that the proposed algorithms enable faster training on new tasks through collecting and transferring the knowledge learned from preceding tasks.

One of the core questions in lifelong learning setting is how tasks should be related for transfer to be effective, and in this paper we are making the HiP-MDP assumption, that is, a family of tasks is generated by varying a latent task parameter vector drawn for each task according to the world-model distribution. The relationship between different tasks can be modeled in different ways in lifelong RL and this problem setting will affect the algorithm's performance. However, we do believe our method can help the agent transfer knowledge for tasks that are in other ways related, as we show in the results on MT10 and MT50, which are not typical HiP-MDP settings.

The overall lifelong learning performance of our proposed algorithm is largely limited by the performance of the single-task model-based baseline. We show the advantages of model-based lifelong learning methods over model-free methods when the single-task baselines' performance are close. In practice, it's possible that the model-free lifelong RL methods still outperforms model-based methods simply because the model-free single-task baseline is much stronger. Bayesian Neural Networks are more computationally expensive than regular neural networks. The longest forward running time among all our experiments is hopper which took around 96 hours to finish training and evaluation on all the tasks. This is in general acceptable time cost and can be potentially improved with more recent BNN techniques.

## Acknowledgement

The authors thank the members of Brown bigAI for discussions and helpful feedback, and the anonymous reviewers for valuable feedback that improved the paper substantially. This research was supported by the NSF under grant #1955361 and CAREER award #1844960, and the DARPA Lifelong Learning Machines program under grant #FA8750-18-2-0117. The U.S. Government is authorised to reproduce and distribute reprints for Governmental purposes notwithstanding any copyright notation thereon. The content is solely the responsibility of the authors and does not necessarily represent the official views of the NSF or DARPA. This research was conducted using computational resources and services at the Center for Computation and Visualization, Brown University.

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
