# OpenReview forum: "Model-based Lifelong Reinforcement Learning with Bayesian Exploration"
_NeurIPS.cc/2022/Conference — NeurIPS 2022 Accept_

### Official Review · Reviewer_E1cW · 2022-07-10

**Rating:** 6
**Confidence:** 4
**Soundness:** 3 good
**Presentation:** 3 good
**Contribution:** 3 good

**Summary:**

* The authors tackle the problem of lifelong learning in RL and propose to use hierarchical Bayesian modeling to achieve transfer between tasks over the course of learning.
* In particular a world model will be used alongside task specific modeling to extract common and specific structure in existing tasks to model new tasks.
* This facilitates modeling new tasks by using the between-task posterior as a within-task prior for newly sampled tasks.
* Tasks are related and sampled sequentially throughout learning and generally depend on a task latent to define specific task features.  Tasks are modeled with HiP-MDPs.
* The authors define forward transfer and backward transfer across tasks by using prior and posterior data to improve the performance across the set of sampled tasks.
* They next derive a performance bound that shows that fewer samples are required to train specific task models as the world model converges on the underlying distribution of environment dynamics.
* The authors further develop Variational Baysian exploration for Lifelong RL (VBLRL) for better scale by using variational inference to approximate a world model using Bayesian Neural Networks.  Both forward and backward (previously encountered tasks) task transfer is implemented to boost performance.
* The authors experiment with this approach on MuJoCo and Meta-world tasks against baselines in lifelong learning and demonstrate state of the art results.

**Claims**:

 1. Demonstrate better forward and backward transfer performance against state of the art lifelong RL algorithms.
 2. Bayesian modeling across tasks that improves sample efficiency.



**Questions:**


* Does forward and backward transfer always result in monotonically increasing performance?  Are there cases where the transfer is counter-productive to performance improvement? Finally, if so, under what conditions does this typically occur?
* You show that sample efficiency improves in the task specific regime when the world model approaches the environment distribution, however is it always the case that the world model improves in this sense?  Is there reason to believe that this will be very limited for some environments?
* Was any study undertaken of the number of particles and what effect this has on posterior sampling and learning convergence?  What impact does this have on the computational complexity?


**Limitations:**

I couldn't see much discussion about limitations.

**Strengths And Weaknesses:**

**Strengths**

* Lifelong RL is an active area of research which presents a number of challenges that will require scalable approaches to overcome.  The authors target a task modelling approach that aims to address this through learning of the underlying dynamics with hierarchical Bayesian methods.
* The authors rigorously demonstrate that task models can be trained with fewer samples as the world model posterior approaches the true underlying dynamics of the tasks environment.
* The authors cover prior work in using Bayesian methods in RL, lifelong RL, and Hip-MDP processes in their related work and provide some background to help contextualise their work.
* Ablation shows BNNs and backward transfer improved performance.
* Results look overall good and VBLRL improves over comparable baselines.  Overall the authors seemed to have satisfied the claims made, demonstrating that their VBLRL approach improves sample efficiency while attaining state of the art performance over a number of baselines.


**Weaknesses**

* There seem to be some missing references in sectionThere is no reference for VBLRL in section 1 and I could not find a reference for "Bayesian Exploration for Lifelong Reinforcement Learning" Fu et al. 2021.  There should be a reference here to prior work.  In the same section BNNs are referenced but no work is cited nor are they really explained.  In section 2, some explanation for both of these approaches would be helpful.
* While the authors demonstrate that the approach is effective the novelty of it seems somewhat limited.  However, the authors provide some theoretical justification.
* Section 4.2 seems like it could use more detail.  For example, figure 1 is fairly high level and doesn't seem to capture in a straightforward way the dynamics of the model learning across tasks.  The algorithms which explain the VBLRL in detail are in the appendix. I feel this section (and 4.3) are meant to really define the unified approach yet I didn't come away with a clear understanding of how back transfer and VBLRL are working here.

---

> ### Author Response · Authors · 2022-08-02
> **Initial Response to Reviewer E1cW - Part 1**
>
> Thank you for the very positive comments and thoughtful suggestions. We address the reviewer’s concerns below.
>
> 1. There is no reference for VBLRL in section 1 and I could not find a reference for "Bayesian Exploration for Lifelong Reinforcement Learning" Fu et al. 2021. There should be a reference here to prior work. In the same section BNNs are referenced but no work is cited nor are they really explained.
>
> A: We did not include a reference for "Bayesian Exploration for Lifelong Reinforcement Learning" Fu et al. 2021 for a very special reason. We have left a message to the Area Chair about this. We have added references to BNNs in the updated version of our paper as the review suggests, and the VBLRL refers to Variational Bayesian exploration for Lifelong RL.
>
> 2. Section 4.2 seems like it could use more detail. For example, figure 1 is fairly high level and doesn't seem to capture in a straightforward way the dynamics of the model learning across tasks. The algorithms which explain the VBLRL in detail are in the appendix. I feel this section (and 4.3) are meant to really define the unified approach yet I didn't come away with a clear understanding of how back transfer and VBLRL are working here.
>
> A: We have added more details about VBLRL as the reviewer suggested. Due to space limits, we put the algorithm pseudocode for VBLRL and VBLRL backward transfer along with more details in Appendix A. For Figure 1, the outer loop (yellow rectangle) stands for the loop of training world-model BNN across tasks (noted at the bottom-right corner), the inner loop (cyan rectangle) stands for the training of the task-specific BNN within each specific task. The task-specific BNN is trained with data only from the current task while the World-model BNN is trained with data from all previously encountered tasks. Details about our BNN structure can be found in Appendix B.
>
> 3. You show that sample efficiency improves in the task specific regime when the world model approaches the environment distribution, however is it always the case that the world model improves in this sense? Is there reason to believe that this will be very limited for some environments?
>
> A: We appreciate the reviewer for pointing out this possibility. The reviewer is correct: our theoretical result that the sample efficiency improves in the task-specific regime depends on the assumption that the world model improves and approaches the true underlying distribution. We would like to point out that the relationship between different tasks can be modeled in different ways in lifelong RL and this problem setting will affect the algorithm’s performance. In this paper, we consider lifelong RL tasks that can be modeled as a HiP-MDP, where a family of tasks is generated by varying a latent task parameter ω drawn for each task according to the world-model distribution. So we believe it’s a reasonable assumption that the world-model distribution over ω will get more and more accurate as the agent encounters more and more tasks (also demonstrated by our experiments). However, for example in cases where the tasks are completely different, the world model will not be improved. We expect overall performance of the agent to improve somewhat, but to eventually reach a performance limit due to model mis-specification. Due to space limit, we include this in the new limitation section in Appendix Q  in response to the review.
>
> 4. Does forward and backward transfer always result in monotonically increasing performance? Are there cases where the transfer is counter-productive to performance improvement? Finally, if so, under what conditions does this typically occur?
>
> A: Following the answer to the last question, when the tasks are entirely different, negative transfer is likely to happen as there’s no common knowledge to be transferred from other tasks. However, we believe hardly any transfer learning methods can really work between two completely different tasks. The core question in lifelong learning setting is how tasks should be related for transfer to be effective, and here we are making the HiP-MDP (Doshi-Velez and Konidaris, 2016) assumption.  In short, our method generally works for HiP-MDP settings, for other categories of relationship between tasks, the forward and backward transfer performance depends on how much common structure the world-model can leverage from different tasks. We presented in Appendix H that for tasks with different kinds of variations on MetaWorld (MT10, MT50),  our method is still able to achieve reasonable performance, though the problem setting is a little out-of-scope for the HiP-MDP setting we are considering in this paper. Note that MT10 and MT50 consist of 10 and 50 different categories of manipulation tasks, e.g., task 1: reach, task 2: push, task 3: drawer close, task 4: button press etc.

---

> ### Author Response · Authors · 2022-08-02
> **Initial Response to Reviewer E1cW - Part 2**
>
> 5. Was any study undertaken of the number of particles and what effect this has on posterior sampling and learning convergence? What impact does this have on computational complexity?
>
> A: We have run ablation studies on the number of particles in response to the review and included the results in Appendix O of the updated version of our submission. The agent performance is close when the number of particles is around 50. The computational complexity drops as we use fewer particles. Note that in this paper for simplicity we use CEM planning as the single-task model-based baseline. As shown in Appendix G, the baseline’s single-task performance still has space to improve so the overall lifelong learning performance may become even better if we replace CEM planning with a better model-based RL algorithm, like Dreamer (Hafner et al.), which may not involve the choice of the number of particles.

---

### Official Review · Reviewer_thFd · 2022-07-11

**Rating:** 6
**Confidence:** 4
**Soundness:** 3 good
**Presentation:** 3 good
**Contribution:** 3 good

**Summary:**

This paper proposes that to more effectively handle life-long reinforcement learning settings, where tasks are presented sequentially, one can use model-based techniques with model uncertainty measured globally across tasks and locally within tasks. By tracking model uncertainty and using planning over stochastic models, the method can better tackle both forward transfer (better performance on future tasks after experiencing previous tasks) and backward transfer (better performance on previous tasks after observing new information from future tasks). The paper demonstrates these results in two continuous, life-long reinforcement learning settings: a suite of Mujoco tasks, and Meta-World, comparing to other model-free techniques as well as approaches that do not model both global and local uncertainty.

**Questions:**

* Can the authors elaborate on how the approach compares to other model-based techniques for RL from meta-learning? Are the technical differences important for getting the approach to work in the life-long setting?
* Can the authors clarify what they believe the main message of the paper is? Is it specifically about the presented method, more generally about the importance of multiple types of uncertainty, or about model-based vs. model-free approaches to life-long reinforcement learning?
* Can the authors discuss how the method would fare in settings that do not have obvious, simple ways in which the tasks differ for life-long RL? What would happen if the presented tasks actually share nothing in common? Would the method successfully avoid catastrophic forgetting?No ethical limitations to consider here. The authors could go into more detail on the technical limitations of the approach.


**Limitations:**

No ethical limitations to consider here. The authors could go into more detail on the technical limitations of the approach.

**Strengths And Weaknesses:**

Strengths
* (Significance) life-long RL is an increasingly critical setting, so paper is addressing an important topic. Furthermore, doing so in a model-based way as this paper demonstrates clearly has advantages over model-free methods, even when the planner used is exceptionally simple, which should encourage people to consider whether model-based methods may be more sensible for their setting.
* (Originality) The method is somewhat novel in estimating both global and local uncertainty within models of transitions using Bayesian Neural Networks. While the overall concept is similar to Nagabandi et al (2018) which models the global and local uncertainty using a MAML-like approach, this paper shows that such a technique is a better choice for the life-long learning setting than alternative life-long RL methods.
* (Clarity) The paper is clearly written. It was straightforward to follow the method and the figures were mostly well explained. As I mention below, I think some of the proofs could be moved to the appendix to make more space for results in the main body of the text.

Weaknesses
* (Quality / Clarity) The choice of baselines seemed somewhat odd, and made the overall message of the paper a bit muddled. In my reading, the most important aspect of the method is to model two layers of uncertainty in the MDPs: globally across tasks, and locally within tasks. The most relevant baselines for this question seem like those that also apply model-based methods to the problem of multi-task learning – for example Nagabandi et al, 2018. I understand their focus was different, but the technique seems very similar and it would help to have better contextualization for why it is different technically, and whether those technical differences matter. As it stands, it was not clear to me if the paper’s main message was an argument for the specific method introduced, or about using model-based methods with planning as a more stable approach to life-long reinforcement learning in general (and the multiple levels of uncertainty just being critical to get this to work). In either case, presenting baselines from both model-based and model-free techniques (and carefully signposting these) would help clarify the message and interpretations of the results.
* (Quality / Clarity) The number of experiments was reasonably small – only one page of results relative to 6 pages of method description. I think some of the proofs/lemmas could be safely moved to the supplement to make more space for results that better show advantages of multi-level uncertainty model-based vs. model-free for lifelong settings and leave more room for discussion around why multiple types of uncertainty are important empirically.
* (Originality) While I am not aware of anyone applying exactly this kind of approach to the continual RL setting, the method feels similar conceptually to several approaches from meta-learning that could also be applied to a life-long setting (but are not included as baselines in this paper).
* (Significance) I am not sure if this will impact how people do lifelong reinforcement learning, as it’s unclear how computationally expensive BNNs are, and whether the approach would work for settings in which the tasks cannot be clearly mapped onto a single parameter which is varying

---

> ### Author Response · Authors · 2022-08-02
> **Initial Response to Reviewer thFd - Part 1**
>
> Thank you for the positive comments and thoughtful suggestions. We address the reviewer’s concerns below.
>
> 1. While I am not aware of anyone applying exactly this kind of approach to the continual RL setting, the method feels similar conceptually to several approaches from meta-learning that could also be applied to a life-long setting (but are not included as baselines in this paper). Can the authors elaborate on how the approach compares to other model-based techniques for RL from meta-learning? Are the technical differences important for getting the approach to work in the life-long setting?
>
> A: The lifelong RL problem setting we consider here is different from meta-learning with continual adaptation settings, in particular compared with “DEEP ONLINE LEARNING VIA META-LEARNING: CONTINUAL ADAPTATION FOR MODEL-BASED RL” Nagabandi et al, 2018. The key technical differences are:
>
> a.  In the lifelong RL settings we consider, tasks are i.i.d sampled from a fixed distribution and arrive one at a time. The agent trains in a sequential manner, as it can only interact with the current task —- the previous and future tasks are both not available. Our evaluation for forward transfer happens each time the agent encounters a new task from the very beginning as the training starts for the first task and proceeds with the training process. By contrast, meta-learning assumes there are a large number of tasks available at the beginning of training and the agent may train a meta-policy by interacting with all these tasks together initially, instead of in a sequential manner. In standard meta-RL settings, the evaluation process happens after the agent finishes training on multiple tasks by sampling several new tasks from the same task distribution and measuring the agent’s performance on them, also not in a sequential manner like in lifelong RL. Thus, we did not empirically compare to standard (model-based) meta-RL approaches as both training and evaluation procedures and objectives are different.
>
> b. In meta learning with continual adaptation like in Nagabandi et al, 2018, the test phase does involve continual adaptation to new tasks and further improvement of the policy. However, like other meta-RL algorithms, it is still based on the assumption that the agent first meta-learns a prior over a number of different tasks before the sequential adaptation process begins. And the continual learning begins when the agent has learned a prior model over a distribution of related tasks. This is also an interesting setting in practice but is different from what we consider here. In our lifelong RL setting, the agent begins with no prior knowledge of the world/tasks, and receives one task at a time with a limited number of iterations for interacting with the task. Moreover, the prior model learned using a MAML-like approach can already perform very well given that a large number of tasks is available during pretraining and the number of sampling iterations is also not limited, while in our case at the earlier stage the world-model that only trains on limited data from a few tasks is highly unlikely to be sufficiently accurate. This is also one of the main reasons that we only use it as an initialization of the task-specific BNN model for each new task, and only use the task-specific model to interact with the new task and do normal training after that. In the theoretical analysis, we demonstrate that the sample complexity of learning on a new task will decrease accordingly as the world model is closer to the true underlying distribution using this kind of initialization.
>
> c. Besides, in our lifelong RL setting, the agent cannot go back to previously encountered tasks and collect new samples from them except for the evaluation phase. This difference also leads to another important problem in lifelong RL that does not usually exist in meta-learning settings – backward transfer. The agent’s performance on previously encountered tasks (especially those encountered at the earlier stage of training) may be bad because the agent can only interact with each task for only a limited number of episodes and the model is inaccurate at that time. So performance has the potential to be further improved (positive backward transfer) as the agent has learned over more and more tasks. In Section 4.3, we proposed a method that can achieve positive backward transfer based on our algorithm’s measurement of different categories of uncertainty. Our separate training of the world model and task-specific model during the forward training process also enables us to combine them to get a better backward transfer performance based on their confidence, which can be relatively easily calculated thanks to the use of BNN.

---

> > ### Comment · Reviewer_thFd · 2022-08-07
> > **Thanks for clarifications**
> >
> > Thanks to the authors for their clear and detailed response. I agree the settings are different, and the authors articulated this clearly. I believe I may have been unclear with my question on this particular point -- it's not that I think meta-RL and lifelong-RL are the same, but rather curious as to how meta-RL approaches would fare in these lifelong learning settings given similarities in the methods (inspired by insights from "Reconciling meta-learning and continual learning with online mixtures of tasks", Jerfel et al 2019).
> >
> > > while in our case at the earlier stage the world-model that only trains on limited data from a few tasks is highly unlikely to be sufficiently accurate. This is also one of the main reasons that we only use it as an initialization of the task-specific BNN model for each new task, and only use the task-specific model to interact with the new task and do normal training after that. In the theoretical analysis, we demonstrate that the sample complexity of learning on a new task will decrease accordingly as the world model is closer to the true underlying distribution using this kind of initialization.
> >
> > My question about this was also inspired by the ablation study performed which seems to suggest that while the whole method improves the mean of the return across tasks, the world model-only ablation is sufficient to be within statistical significance of the full method (Figure 3). Could the authors comment on this further?

---

> > > ### Author Response · Authors · 2022-08-07
> > > **Response to post-rebuttal comments from Reviewer thFd**
> > >
> > > We appreciate the reviewer’s response very much!
> > >
> > > Firstly, because meta-RL and lifelong-RL algorithms are for different settings, we don’t think they can be directly compared. We agree that only using the world model for all the tasks in our case is conceptually very similar to model-based meta-RL. We included this baseline in experiments on all the domains under the name “T-HiP-MDP'', where the proposed method VBLRL achieves significantly better forward and backward transfer performance in most domains (see Table 1, Figure 2, Figure 5). Note that the difference between T-HiP-MDP and VBLRL is that T-HiP-MDP only maintains one single model (also modeled with Bayesian Neural Networks) that is continually learned over the current task and samples collected from previous tasks. So we believe the T-HiP-MDP baseline can also show how a meta-RL-like approach performs in lifelong learning settings.
> > >
> > > The ablation study shown in Figure 3 is only for backward transfer evaluation. The difference between the world-model-only backward transfer baseline here and the T-HiP-MDP backward baseline in Table 1 is that the world model in this ablation study is also learned with VBLRL. That is, the data on which the world model is trained is collected by VBLRL’s task-specific model’s policy, while for T-HiP-MDP, the data is collected by the world model’s policy. This leads to different quality of exploration — the task-specific model’s policy only cares for the current task and is prone to collect useful data more efficiently, while the world model’s policy aims to achieve an average good performance on all previous encountered tasks. This is also implied by the results in Table 1 if we only compare the “start” performance of VBLRL and HiP-MDP. On a new task, both of the methods are initialized with world model but VBLRL has better performance, indicating that the world-model learned by VBLRL are better because of the different exploration quality. However, no matter how well the world model performs, for a specific task only using the world-model to do backward transfer will miss task-specific information, which leads to overall lower average return compared to combining it with task-specific model as shown in Figure 3. Moreover, if we compare VBLRL's "start" and "train" performance in Table 1, we can see that after initializing with the world-model, the agent's forward transfer performance can still be well improved by training the task-specific model.

---

> ### Author Response · Authors · 2022-08-02
> **Initial Response to Reviewer thFd - Part 2**
>
> d. Another difference in the problem setting of our work compared with Nagabandi et al., 2018 is that in our setting the task indicator (whether the task the agent is currently interacting with has recently switched) is known. This is the same setting addressed in LPG-FTW (Mendez et al., NeurIPS 2020). However, Nagabandi et al. focus on the setting that the task indicator is not available, so it requires the algorithm to focus on training the agent to be able to identify the current task.
>
> 2. Can the authors clarify what they believe the main message of the paper is?
>
> A：In this paper, we focus on the lifelong reinforcement learning problem, where the relationship between different tasks is modeled as HiP-MDP. The main message of the paper is the proposed lifelong RL algorithm (VBLRL), which is based on the following two main intuitions: 1. transferring the transition model instead of policy/value function leads to more efficient use of the data in lifelong RL. Our theoretical results show a way to transfer the world-model that can lead to sample efficiency improvement. Our empirical results demonstrate the intuition by comparing the performance of VBLRL with LPG-FTW and EWC. 2. Separately representing the world-model that captures the uncertainty globally across tasks and the task-specific model that captures the uncertainty within specific tasks using Bayesian neural networks helps the agent get better forward and backward transfer performance. Our empirical results demonstrate this by comparing VBLRL with T-HiP-MDP and Single-task MBRL, as well as an ablation study about using Bayesian neural networks or not. All the results together show the advantages of our proposed algorithm (VBLRL) in such lifelong RL settings.
>
> 3. The reasons for the choice of baselines in the experiments:
>
> A: Model free methods: 1. LPG-FTW (Mendez et al., NeurIPS 2020) as mentioned above is the state-of-the-art lifelong RL algorithm. 2. EWC (Kirkpatrick et al.) is another recently proposed standard lifelong RL algorithm that achieves comparable performance with LPG-FTW as shown in the latter paper. The single-task model-based baseline (CEM-planning) we used achieves similar performance to the model-free baseline used by LPG-FTW and EWC.  Comparing our method to these two baselines highlights the advantage of the proposed model-based techniques for sample efficiency in lifelong RL setting.
>
> Model-based methods: 1. T-HiP-MDP (Killian et al.) is exactly a model-based algorithm and for a fair comparison we further replaced the DDQN algorithm used in the original paper with CEM planning as is used in our method. This baseline is conceptually similar to model-based meta-RL, as the version we implemented only maintains one single model (also modeled with Bayesian Neural Networks) that is continually learned over the current task and samples collected from previous tasks. It is expected to generate policies that can perform well on all the tasks in the same distribution. Comparing this baseline with our algorithm also demonstrates the importance of estimating local uncertainty as this baseline does not have a task-specific model like our method.
> 2. Single-task MBRL. This is a simpler version of our algorithm that does not use the world model to initialize the task-specific model, and trains the task-specific model from scratch each time encountering a new task. We are not aware of any other existing model-based lifelong deep RL algorithms at the time of submission.
>
>
> 4. The number of experiments was reasonably small:
>
> A: We believe we have conducted sufficient experiments to support our claims. In the two-page experiments section in the main text, we show the results on six different challenging mujoco domains, results on two different meta-world domains and ablation study on two components of VBLRL. Because we use a table to conclude the results on six different mujoco domains, it does look very dense. But in that table we actually include each algorithm on each domain’s performance at different training stage (start, train, back) to demonstrate the agent’s performance of forward transfer and backward transfer, and compared to four different baselines. Besides the experiments shown in the main text, we include experimental results on another two domains in MetaWorld in Appendix H, single task baseline comparison in Appendix I, full lifelong RL comparison in Appendix J, as well as experimental results on two additional simpler domain (GridWorld item searching, Box-jumping) to demonstrate the performance of the finite MDP version of our proposed algorithm in Appendix L and M. We also include new experiment results for the ablation study of the number of particles in Appendix O. We moved some proof sketches to the appendix as the reviewer suggests and we believe the remaining parts are important for understanding the paper.

---

> > ### Comment · Reviewer_thFd · 2022-08-07
> > **Further response**
> >
> > > d. Another difference in the problem setting of our work compared with Nagabandi et al., 2018 is that in our setting the task indicator (whether the task the agent is currently interacting with has recently switched) is known. This is the same setting addressed in LPG-FTW (Mendez et al., NeurIPS 2020). However, Nagabandi et al. focus on the setting that the task indicator is not available, so it requires the algorithm to focus on training the agent to be able to identify the current task.
> >
> > That's very reasonable, but it doesn't seem like a limitation of another method if they can perform well even without the task indicator. I agree with your previous assessment of the task setting being different though, and will defer to that other comment for further discussion on this point.
> >
> > Thanks for clarifying the choice of baselines -- some extra discussion of this in the paper would be helpful, as I found this very illuminating.
> >
> > Thank you for pointing out the experiments and pointing towards the appendix. I agree these are sufficient to demonstrate the value of the method. One minor request: could you please only bold the numbers for the method when they are statistically significantly better than the baselines? This helps with interpretation of the results.

---

> ### Author Response · Authors · 2022-08-02
> **Initial Response to Reviewer thFd - Part 3**
>
> 5. Can the authors discuss how the method would fare in settings that do not have obvious, simple ways in which the tasks differ for life-long RL? What would happen if the presented tasks actually share nothing in common? Would the method successfully avoid catastrophic forgetting?
>
> A: Firstly, we believe hardly any transfer learning methods can really work between two completely different tasks. One of the core questions in lifelong learning setting is how tasks should be related for transfer to be effective, and here we are making the HiP-MDP (Doshi-Velez and Konidaris, 2016) assumption.  All the experiment domains (Gym mujoco, MetaWorld reach/reachwall, gridworld) considered in this paper are in the HiP-MDP setting. Note that HiP-MDP assumption is also made in the experiments of many recent papers about transferring knowledge in RL (VARIBAD: A VERY GOOD METHOD FOR BAYES-ADAPTIVE DEEP RL VIA META-LEARNING, ICLR 2020; MetaCURE: Meta Reinforcement Learning with Empowerment-Driven Exploration, ICML 2021).The performance of a lifelong RL algorithm can be largely determined by what relationship is between different tasks. If the tasks are entirely different, negative transfer is likely to happen as there’s no common knowledge to be transferred from other tasks.  Due to space limit, we include this in the new limitation section in Appendix Q in response to the review. We presented in Appendix H that for tasks with different kinds of variations on MetaWorld (MT10, MT50),  our method is still able to achieve reasonable performance, though the problem setting is a little out-of-scope for the HiP-MDP setting we are considering in this paper. Note that MT10 and MT50 consist of 10 and 50 different categories of manipulation tasks, , e.g., task 1: reach, task 2: push, task 3: drawer close, task 4: button press etc. This is also a response to the case “settings in which the tasks cannot be clearly mapped onto a single parameter which is varying”  the reviewer mentioned before, although we want to emphasize here in the previous evaluated domains the tasks are mapped onto a parameter vector which has more than one dimension.

---

> > ### Comment · Reviewer_thFd · 2022-08-07
> > **Thanks for pointing to these experiments**
> >
> > Yes, this is a great experiment to point this out. However, I am confused -- I thought Meta-World was not a lifelong reinforcement learning domain, but rather a multi-task and meta-learning benchmark, which (as pointed out over several of the author's comments) is very different from the lifelong RL setting being considered. Could the authors explain how they change Meta-World to be a life-long reinforcement learning task, and where this is explained in the paper?

---

> > > ### Author Response · Authors · 2022-08-07
> > > **Response to post-rebuttal comments from Reviewer thFd**
> > >
> > > We thank the reviewer for pointing this out. We understand the reviewer’s confusion as we didn’t make this clear enough. MT10 and MT50 are built-in multi-task benchmarks in MetaWorld, which consist of 10/50 different categories of manipulation tasks. We modified these two domains to be lifelong learning settings: instead of getting all 10/50 tasks at the same time at the beginning, we let them arrive sequentially. For instance, the first task is “reach” and we let the agent interact with it for 150 iterations, and then switch to “push” and the agent cannot collect data from “reach” anymore, after another 150 iterations the task changes to “pick-place”, etc. We evaluate the agent’s performance on all the tasks like in gym mujoco domains. We included this description in Appendix H in response to the review.

---

> > > > ### Comment · Reviewer_thFd · 2022-08-07
> > > > **Thanks for the clarifications - I have raised my score**
> > > >
> > > > Thank you for the clarifications and the detailed replies. I have raised my score accordingly across several of the metrics.

---

> > > > > ### Author Response · Authors · 2022-08-07
> > > > > **Thank you!**
> > > > >
> > > > > Thank you very much for your helpful feedback and for increasing your score!

---

### Official Review · Reviewer_osbM · 2022-07-12

**Rating:** 7
**Confidence:** 3
**Soundness:** 3 good
**Presentation:** 3 good
**Contribution:** 2 fair

**Summary:**

This paper proposes an approach for lifelong learning for a set of parametrically varying tasks, by maintaining a joint world model across all tasks encountered, as well as task-specific models. In the absence of task-specific data, the prior of the task-specific models is set to the world model posterior. The paper includes theoretical analysis that shows that this leads to increased sample efficiency on the task, and also includes empirical evaluation on a set of gym tasks with parametric variation (eg: gravity, mass of body parts), as well as meta-world reaching.


**Questions:**

-> How much sample efficiency benefit does using the joint world model yield, as opposed to re-training from scratch with only the task-specific model?

-> Does the approach scale to settings with more meaningful variation across tasks (eg- reach, push, pickplace) in metaworld?

**Limitations:**

Limitation are not adequately discussed. As described before, the primary limitation is the quality of experimental domains considered.

**Strengths And Weaknesses:**

**Update after Author Response**

I thank the authors for their detailed response. The authors have sufficiently addressed all of my concerns (described in detail below). In light of these clarifications, the  new experimental evidence included and given that the approach does seem original and intuitive, and the problem considered is of importance to the community (as noted in my original review), I am updating my score. With the new results, I think the paper will be of great relevance to the lifelong RL community.


-> How much does the world model prior help? The authors clarify that this is actually included in the paper, though not clearly described in the writing (which can be easily amended for the camera ready version). Further, the authors add new experiments (appendix L) to directly compare to a version of their method using just a fixed prior, to highlight the benefit of using the world model prior.

->  Can the method handle semantically varying tasks? Lifelong learning approaches will need to handle this setting when they're deployed for problems of interest in robotics. The initial experiments were run on parametrically varying tasks (different gravity, body mass etc) or transferring from reaching to reaching with a wall. I appreciate the effort of the authors in testing their approach on the much more challenging MT-10 and MT-50 benchmarks (Figure 5, Appendix H), which have actually different tasks (eg: opening a boxes/closing drawers/opening windows etc). These results make the claims of the paper \emph{significantly} strengthen the claims of the paper, and demonstrate the effectiveness of the approach in comparison to current methods.

-> How does the method compare to robust policy ? The authors clarify that this considered in the paper that originally introduced the tasks considered in the initial experiments. Further, this concern is sufficiently addressed with the MT-10/MT-50 experiments. The authors also add discussion related to this in appendix Q.

** Original Review **

Significance  -

-> Lifelong learning is an important challenge for RL, as learning from scratch for each new task is infeasible, especially for robotics. Proposed solutions are more likely to be adopted by the community at large if they show signs of progress on tasks closer to the actual problem setting, i.e a stream of semantically different tasks that are encountered over an agent’s lifetime. The main results in the paper are presented on a set of gym tasks (cheetah, walker) with very simple parametric variations (changing gravity, changing body mass etc). For such domains, since access to all previous data is assumed, one could train a single robust model-free policy that can perform quite well, using algorithms like SAC. Furthermore, the metaworld suite includes semantically different tasks (reaching, pushing, pick-place etc) and a better evaluation of the algorithm would include evaluating the approach on these variations. Without this it is difficult to understand whether the approach is actually applicable to the problems faced by RL for deployment.

-> More broadly, part of the issue stems from considering the set of tasks to only have parametric variation. While this restriction might provide added structure to help in theoretical analysis, it is worth considering if the same general idea (joint world model which is used to initialize task-specific models) can also be used for more general sets of tasks, that do not vary in terms of an interpretable parameter. This would open up the approach for evaluation on semantically varying tasks, as even in this case there are similarities to be shared across tasks, such as physics.

Quality -

->  The empirical results don’t include sample efficiency comparison of the approach without using the world model to initialize the task-specific models. This seems important as it’s one of the central claims of the paper. Furthermore, as described before the experimental domains are too far removed from settings of interest for the deployment of RL for lifelong learning.

-> The probabilistic formulation of the individual task models and the joint world model and the corresponding update schemes are sound  and intuitive to follow. The theoretical result that initializing the task specific prior to the joint world model posterior leads to sample efficiency gains also seems intuitive, given that the joint model is trained across tasks.

Originality -

->  The proposed approach using world models for continual learning does seem new, and can potentially be very impactful since transition dynamics should be preserved across tasks, and this is exactly what the models learn. Further, the authors provide a probabilistic framework for learning on a particular task while leveraging prior knowledge from a joint model, and provide theoretical analysis for the advantage in doing so. The related work section is adequate and includes most prior work.

Clarity -

-> The paper is clearly written and not too difficult to follow, and includes detailed algorithms that are sufficient for reproducing the method.

---

> ### Author Response · Authors · 2022-08-02
> **Initial Response to Reviewer osbM- Part 1**
>
> Thank you for the thoughtful review; we believe we can resolve your concerns.
>
> 1. The empirical results don’t include sample efficiency comparison of the approach without using the world model to initialize the task-specific models.
>
> A: We agree that such a comparison is central to the paper and included it under the name “Single-task MBRL”. It is present in all six domains of mujoco and two domains of MetaWorld. We understand the reviewer’s confusion as we did not make the role of this baseline clear in the paper---we apologize. In all 8 domains, that method was hardly able to learn a useful policy within the limited interactions with each task, except in walker-gravity and walker-bodyparts. To make our claim stronger, in Appendix L we also compared the finite-MDP version of our algorithm (BLRL) with BOSS where BOSS does not have a world-model prior (no knowledge transfer). This comparison supports our theoretical results in Proposition 4.4.
>
> 2. The main results in the paper are presented on a set of gym tasks (cheetah, walker) with very simple parametric variations (changing gravity, changing body mass etc). For such domains, since access to all previous data is assumed, one could train a single robust model-free policy that can perform quite well, using algorithms like SAC.
>
> A: Note that we adopted the six gym mujoco benchmarks directly from a well regarded lifelong RL paper, LPG-FTW (Lifelong Policy Gradient Learning of Factored Policies for Faster Training Without Forgetting. Mendez et al., NeurIPS 2020). Appendix C of that paper showed that these tasks are diverse enough that a single policy does not work across various tasks. We choose to use a very simple single-task model-based baseline – CEM planning to be comparable with the model-free baseline used in LPG-FTW (single-task performance comparison shown in Appendix G). We don’t see an obvious way to integrate SAC or other recent better model-free baselines into LPG-FTW as the method is built upon explicit calculation of policy gradient. The performance of our method, however, can be potentially further improved by replacing CEM with a better model-based RL algorithm. But comparing it with LPG-FTW/EWC then would not be fair: it will not be clear whether the empirical benefit is brought by the single-task baseline or the main components we proposed in this paper. However, the flip side is that the overall lifelong learning performance of our proposed algorithm is also limited by the performance of the single-task model-based baseline —- in practice it's possible that the model-free lifelong RL methods still outperforms model-based methods when the model-free single-task baseline is much stronger.  Due to space limit, we include this in the new limitation section in Appendix Q in response to the review.

---

> ### Author Response · Authors · 2022-08-02
> **Initial Response to Reviewer osbM- Part 2**
>
> 3. Furthermore, the metaworld suite includes semantically different tasks (reaching, pushing, pick-place etc) and a better evaluation of the algorithm would include evaluating the approach on these variations. Without this it is difficult to understand whether the approach is actually applicable to the problems faced by RL for deployment. It is worth considering if the same general idea (joint world model which is used to initialize task-specific models) can also be used for more general sets of tasks, that do not vary in terms of an interpretable parameter. Does the approach scale to settings with more meaningful variation across tasks (eg- reach, push, pickplace) in metaworld?
>
> A: Firstly, we believe that transfer learning is not suited to completely different tasks. One of the core questions in lifelong learning setting is how tasks should be related for transfer to be effective, and here we are making the HiP-MDP (Doshi-Velez and Konidaris, 2016) assumption, that is, a family of tasks is generated by varying a latent task parameter vector ω drawn for each task according to the world-model distribution. Of course, the agent is not given that model but must learn it. All the experiment domains (Gym mujoco, MetaWorld reach/reachwall, gridworld) considered in this paper are in the HiP-MDP setting. Note that besides LPG-FTW, which is a representative lifelong RL paper, many recent meta-RL papers also follow this HiP-MDP setting in all the experiments (VARIBAD: A VERY GOOD METHOD FOR BAYES-ADAPTIVE DEEP RL VIA META-LEARNING, ICLR 2020; MetaCURE: Meta Reinforcement Learning with Empowerment-Driven Exploration, ICML 2021). Meta-learning conceptually considers the same problem of transferring knowledge learned from prior tasks to a new task, yet the HiP-MDP setting is also widely used. That said, we do believe our method can help the agent transfer knowledge for tasks that are in other ways related like the cases the reviewer mentioned, though it is out of the scope of the HiP-MDP setting we consider here.
>
> To test whether our idea is applicable to lifelong learning problems with different categories of task relationships, we have added new experiment results on MT10 and MT50 tasks of MetaWorld  in Appendix H in response to the review. MT10 and MT50 consist of 10 and 50 different categories of manipulation tasks, including reach, push, drawer close, button press etc. With the limited number of interactions for each task, our proposed algorithm is still able to achieve reasonably good performance in such settings. In MT10, the total number of tasks (10) is relatively small so the world-model distribution may be far from accurate, but VBLRL still performs better than Single-task MBRL which does not use the world model to initialize the task-specific model. In MT50, the number of tasks is large enough (50) and we can see that VBLRL significantly outperforms the other baselines. This is consistent with our intuition that sample efficiency improves in the task specific regime when the world model approaches the environment distribution. As the single-task model-based RL baseline we used here is relatively simple (CEM planning) to be comparable with the single-task baseline used by LPG-FTW, it is reasonable to expect the lifelong learning performance to be even better after combining our framework with model-based methods like Dreamer (Hafner et al.); we leave such evaluations for future work.

---

> ### Author Response · Authors · 2022-08-08
> **Any follow-ups?**
>
> We want to thank the reviewer once again for helping us improve our paper. We have run experiments on MetaWorld MT10 & MT50 domains as the reviewer suggested. We also made the role of the baseline "Single-task MBRL" clearer in the paper, which is exactly retraining from scratch with only task-specific model as the reviewer asked. Given that the final stage of discussion ends on Aug 9, we’d like to ask if the reviewer has any remaining questions or if there is anything else we can clarify?

---

> ### Comment · Reviewer_osbM · 2022-08-09
> **Review Update**
>
> I thank the authors for their response, I have updated my review.

---

### Meta-Review · Area_Chair_XbkR · 2022-08-28

**Recommendation:** Accept
**Confidence:** Certain

**Metareview:**

Unanimous accept from 3 experienced reviewers with good confidences

Important topic (lifelong RL), clearly explained, well-formulated with a variational and hierarchical Bayesian model, evaluated on a range of relevant experiments in discrete and continuous settings in MuJoCo, and also MetaWorld MT10 & MT50 domains in response to reliever osbM and thFd questions, adapted to the lifelong setting of rolling out tasks sequentially, to include like opening a boxes/closing drawers/opening windows.

**Award:**

No

---

### Decision · Program_Chairs · 2022-09-14

Accept